# Preserved Voluntary Micturition Control despite Early Urinary Diversion in Infancy—A Clue to a New Strategy

**DOI:** 10.3390/children9050600

**Published:** 2022-04-23

**Authors:** Dominika Borselle, Dariusz Patkowski, Katarzyna Kiliś-Pstrusińska, Wojciech Apoznański

**Affiliations:** 1Department of Pediatric Surgery and Urology, Wroclaw Medical University, Borowska 213, 50-556 Wroclaw, Poland; dariusz.patkowski@umw.edu.pl (D.P.); wojciech.apoznanski@umw.edu.pl (W.A.); 2Department of Pediatric Nephrology, Wroclaw Medical University, Borowska 213, 50-556 Wroclaw, Poland; katarzyna.kilis-pstrusinska@umw.edu.pl

**Keywords:** toilet training, urinary diversion, micturition skills, voluntary micturition, urinary tract function

## Abstract

Micturition is an involuntary process based on spinal arcs in infants and children until a defined age. The awareness and voluntary control of voiding depends on specific areas in the central nervous system, especially cortical regions. The cells and connections between these areas develop over time and regulate the voiding process. The ability to maintain continence and to adjust physiological needs to appropriate environmental conditions is considered to be acquired through systematic behavioral education, especially toilet training. The recommendations specify the age at which to start establishing the relevant habits. The purpose of these guidelines is to achieve proper micturition control development and to avoid functional lower urinary tract (LUT) disorders. We present a case of a patient who underwent complete urinary diversion in infancy and reconstruction of the urinary tract eleven years later. For eleven years, she had an empty bladder and no toilet training. After undiversion, she regained full continence in a short space of time. The presence of proper LUT function and a controlled micturition process raises the question of the standard toilet training recommendations’ validity. The aim of our work focuses on the following question: Is toilet training the only way to achieve micturition skills and proper urinary tract function? The history of our patient and the literature reveal that voluntary micturition may develop without stimulating signals of filling from bladder receptors and independently of recommended behavioral education, so toilet training seems to not be necessary.

## 1. Introduction

Micturition is an involuntary process in infants and children until the age of 3–5 years. After this period, it is under voluntary control and is coordinated by the brain, the spinal cord, and the peripheral nervous system (autonomic and somatic) with various neurotransmitters [1,2,3].

In our study, we present the noteworthy case of a patient who underwent complete urinary diversion in early infancy and reconstruction of the urinary tract at the age of eleven. For eleven years, she had an empty bladder and no toilet training. The function of the lower urinary tract (LUT) for urine storage and as a means of urine outflow after this treatment was preserved and consciously controlled. 

The relevant aim of our work is to analyze and verify the reasonableness of toilet training—is it still necessary in order to achieve micturition control development, proper urinary tract function, and to avoid LUT conditions?

## 2. Case Presentation

The patient, a twelve-year-old girl, was admitted to our Department of Pediatric Surgery and Urology to undergo an operation of left ureterocutaneostomy closure. She was diagnosed and underwent surgery in the neonatal and infant periods with congenital complex genitonephrourinary malformation in another department. The analysis of the available clinical data allows us to evaluate the diagnostic and therapeutic process.

The first ultrasonography after birth revealed hypodysplasia of the right kidney, two intra-abdominal cystic structures, and a normal-sized left kidney. She then developed some severe urinary tract infections (UTIs) with an increased creatinine level and features of acute kidney injury treated with antibiotics. The kidney function was diminished. The dynamic renal scintigraphy after successful UTI treatment revealed a significantly decreased function of the right kidney and a prolonged secretory phase of the left kidney. The CT showed the presence of high-radiopacity urine inside one of the abdominal cystic structures. The voiding cystourethrography revealed a left vesicoureteral reflux.

The patient was scheduled for a laparotomy. Intraoperatively, the diagnoses of vaginal duplication with an obstructed right vagina and ectopic ureteral insertion into the obstructed hemivagina were established. During the procedure, a connection between two parts of the hemivagina was made. The postoperative period was complicated by UTI and kidneys failure. The decision for urinary diversion was taken, so a left terminal ureterocutaneostomy was performed. The ureterocutaneostomy had been functioning properly throughout. The patient grew up and became more aware of her condition, involving being incontinent. 

The scintigraphy performed after surgical treatment revealed a nonfunctioning right kidney and proper function of the left kidney. The creatinine level was elevated to 1.6 mg/dl. In the routine ultrasonography examinations, the left ureter and pyelocaliceal system were not dilated. For ten years, she was left with left ureterocutaneostomy without any surgical treatment. The medical treatment was occasional antimicrobial prophylaxis. 

The cystometry performed at the age of ten revealed stable detrusor function, without involuntary bladder contractions, decreased compliance, and a low bladder capacity. The sensation of bladder filling was proper. After fluid intake, applying voiding was effective without residual urine. The surgery for lower urinary tract reconstruction was preceded by bladder conditioning. It consisted of intravesical infusions of physiological saline in increasing portions using catheter about six months before the operation. Moreover, anticholinergic drugs (oxybutynin 5 mg twice a day) were administered. Both bladder conditioning and pharmacological mechanisms led to the bladder volume increasing to approximately 100 mL, which permitted the risk of performing ureterocutaneostomy closure. 

At the age of eleven, the patient underwent ureterocutaneostomy closure and left ureter reimplantation to the bladder. The postoperative course was complicated by UTI with transient creatinine elevation, treated effectively by antibiotics. The bladder was regularly trained by Foley catheter closure until urinary urgency. After seven days of bladder training, the catheter was removed. The patient has presented a controlled micturition process with a proper sensation of bladder filling and effective detrusor-sphincter coordination. Nowadays, she is usually continent. Early increased urinary frequency has decreased over time with an increase in bladder volume. The kidney function parameters are stable. The cystometry performed after six months from surgery revealed an increase in bladder volume to 250 mL, and after twelve months, it was elevated to 350 mL. The intravesical pressure is within normal limits.

## 3. Discussion

The purpose of our study is to present the phenomenon of controlled micturition and remaining bladder function in a patient who underwent urinary diversion in infancy and reconstruction of the lower urinary tract eleven years later. The history of our patient highlights reflections on the current assumptions and suggests a contrary opinion to toilet training recommendations [4]. The analysis of our patient’s clinical course may distinguish a group of children that achieve micturition habits without training. The patient had lost the ability of voiding physiologically by the bladder and the urethra at the age of two months. As a result, she has never undergone voiding education because she has never needed it. During the period of intensive neural and cortical control development, she had terminal left ureterocutaneostomy and the bladder could not receive appropriate filling signals and send them to higher parts of the nervous system that could stimulate enhancement of the micturition process. The central nervous system was not regulated by afferents because of the “forgotten” bladder [5]. 

In our case, the patient had complex genito- and nephrourinary (CAKUT) malformation and recurrent UTIs that led to the necessity of saving the single left kidney function by conducting terminal ureterocutaneostomy. However, there was no creatinine level decrease within eleven years of observation and with a functioning ureterocutaneostomy. This condition resulted in the impossibility of achieving "dryness” [4]. This significantly influenced the daily functioning of the patient and her parent. It is important to include the psychological aspect of this condition. 

It was difficult for us to explain why the urostomy had not been closed at an earlier stage of her life. The creatinine level remained stable after operation. Perhaps the reason was fear of the single kidney function. The relevance of urinary diversion and maintaining the ureterocutanestomy was based on urological explanations and the presence of UTIs. The current studies indicate that renal damage in children has been found to be more congenital in origin than was previously thought [6]. For example, some randomized controlled trials on vesicoureteral reflux (VUR) and UTIs questioned their conventional clinical significance for renal scarring [7]. Anomalies of the urinary tract are often associated with renal parenchymal changes, typical for renal hypoplasia or dysplasia [8]. Children with CAKUT often have varying degrees of a reduced number of nephrons at birth [9]. Both can lead to chronic kidney disease, so the patients need nephrological control, independently from the underlying etiology of CKD [10]. The creatinine level has not decreased below established value of 1,6 mg/dl within eleven years of a functioning ureterocutaneostomy. The ureterocutaneostomy did not prompt an improvement in kidney function parameters. This proves the congenital origin of chronic kidney disease in our patient’s condition. This justified the decision for ureterocutaneostomy closure. The most important issues that made the decision of surgical treatment possible were the relatively efficient secretory function of the left kidney in scintigraphy, stable kidney function parameters in laboratory tests, no UTIs for a long time, and positive urodynamic examination results. The laboratory data analyses of the presented patient have indicated the stable function of the single kidney during follow-ups. After the lower urinary tract reconstruction, a deterioration in kidney function has not been observed, which indicates that the means of urine output did not affect the preservation of kidney function.

The relevant aim of our work is to analyze and verify the reasonableness of toilet training—is it still necessary in order to achieve micturition control development, proper urinary tract function, and to avoid LUT conditions? Our social environment requires children to be continent at a certain age, which means full control of the storage of urine and emptying of the bladder at an appropriate time and place [4]. In Western Europe culture, voiding education with toilet training is usually administered in children below the age of five. Bloom et al. reported a “toilet-training age” for daytime urinary control at 2.4 +/− 0.6 years [11]. This training involves regular use of the toilet and knowledge about recognizing body signals and communicating physiological needs. It also relies on positive conditioning. The education also involves behavioral training (restriction of late liquid supply and voiding just before falling asleep), wake-up training, bladder wall compliance, and distending exercises (increased daily liquid supply and extension of intervals between voidings) [12]. However, experience suggests that some children do not acquire urination control despite the training. 

Based on our case analysis, we present the phenomenon of voluntary controlled micturition that developed independently of stimulating signals from bladder receptors and behavioral toilet training. This refers to the initial question of our case report and indicates that toilet training may not be the only way to achieve micturition skills and proper lower urinary tract function. 

## 4. Conclusions

Our case report pays attention to the development of lower urinary tract function control. Further studies with a greater number of participants are required to confirm our hypothesis. However, the analysis of a single patient’s history leads to the conclusion that voluntary, conscious micturition, which is under the control of the central nervous system and cortical areas, may develop without stimulating signals of filling from bladder receptors and independently of the recommended behavioral education. Behavioral toilet training seems to not be necessary in order to achieve micturition skills and appropriate urinary tract function, nor does it seem necessary in order to avoid the occurrence of lower urinary tract symptoms.

## Data Availability

Availability of data and materials: the datasets used and/or analyzed during the current study are available from the corresponding author on reasonable request.

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
