# Peer review of "Preserved Voluntary Micturition Control despite Early Urinary Diversion in Infancy—A Clue to a New Strategy"

_children, 2022, doi:10.3390/children9050600_

Round 1

Reviewer 1 Report

The authors present a single case of a 11 year girl with voluntary micturition control over time after surgical reconstruction. The child has 6 months of catheter infusion prior to the surgery and regularly trained for 7 days after surgery. The conclusion was of a changing strategy over toilet training in general

However, I believe that a single case is not sufficient for long - term conclusion of with such changing of strategy. It takes comprehensive research for such conclusion. 

A conclusion for a synergistic approach should be more plausible. 

Author Response

  1. I improved conclusions and a paragraph of discussion which refer to results. I agree that single-case report is not sufficient to such general conclusion and I added corrections.

Thank You for the review.

Dominika Borselle

Reviewer 2 Report

Well written paper. Anticholinergic drug can not increase bladder volume. It's  unclear. Authors should re-write that point.

Author Response

I improved a paragraph of pharmacological treatment. I wrote that both mechanisms together: bladder conditioning and anticholinergic drugs led to bladder volume increase. 

Thank You for the review.

Dominika Borselle

Round 2

Reviewer 1 Report

I believe that the authors have made the necessary changes and I believe that the manuscript is now suitable for publication